# Identification of Naturally Occurring Inhabitants of Vaginal Microbiota in Cows and Determination of Their Antibiotic Sensitivity

**DOI:** 10.3390/vetsci12050423

**Published:** 2025-04-29

**Authors:** Zsóka Várhidi, Viktor Jurkovich, Péter Sátorhelyi, Balázs Erdélyi, Orsolya Palócz, György Csikó

**Affiliations:** 1Department of Animal Hygiene, Herd Health and Mobile Clinic, University of Veterinary Medicine, 1078 Budapest, Hungary; 2Centre for Animal Welfare, University of Veterinary Medicine, 1078 Budapest, Hungary; 3Fermentia Microbiological Research and Development Ltd., 1045 Budapest, Hungary; 4Department of Pharmacology and Toxicology, University of Veterinary Medicine, 1078 Budapest, Hungary; palocz.orsolya@univet.hu (O.P.); csiko.gyorgy@univet.hu (G.C.)

**Keywords:** dairy cattle, vaginal microbiota, antibiotic sensitivity, reproductive health, microbial balance, antimicrobial resistance

## Abstract

Understanding the natural composition of the vaginal microbiota in dairy cows is crucial for maintaining reproductive health and preventing diseases. This study identified the bacterial species present in the vaginal microbiome of healthy dairy cows and assessed their sensitivity to commonly used antibiotics. A total of 54 bacterial species were detected. The antibiotic susceptibility tests revealed that, while some bacteria carried resistance genes, most remained sensitive to multiple antibiotics. These results may contribute to improved herd management strategies by promoting microbiota balance and supporting the responsible use of antibiotics on dairy farms.

## 1. Introduction

The vaginal microbiota is a central element of reproductive health in dairy cattle as a primary defence against external pathogens and ensuring a healthy uterine environment [1,2]. During the periparturient period, the microbial composition of the reproductive tract undergoes dynamic changes driven by physiological, environmental, and management factors [3,4]. Factors such as hormonal fluctuations, calving practices, and dietary management significantly influence the diversity and abundance of vaginal microbiota [5]. For instance, oestrogen levels have been shown to promote the proliferation of specific bacterial species during the oestrous cycle [2,5,6]. Similarly, management practices, including maintaining hygiene during calving, can alter the microbial balance, influencing reproductive outcomes. Breeding strategies and genetic factors may also play a role, as studies reveal breed-dependent variations in microbiota composition [2]. A well-balanced vaginal microbiota has been linked to improved post-calving outcomes, while dysbiosis is frequently associated with uterine diseases, such as endometritis and metritis. Notably, cows that develop uterine diseases often exhibit reduced bacterial diversity in their reproductive tract microbiota compared to their healthy counterparts [2,6]. As ascending infections of the uterus, through the vagina, occur in a significant proportion of freshly calved cows, exploring and supporting the healthy vaginal microbiome may be pivotal in preventing uterine disease [7,8,9,10]. Recent advances in high-throughput sequencing technologies have provided new insights into the composition and diversity of microbial communities within the bovine reproductive tract [11,12]. These findings challenge the conventional view that uterine infections are solely the result of environmental pathogens. Instead, it raises the possibility of the endogenous transfer of microbes from the gut or vaginal niches. For example, healthy cows typically restore their uterine microbiota rapidly post-calving, while those predisposed to uterine disease may exhibit delayed or incomplete microbiota re-establishment [3,4,13].

Researchers have recently begun to focus on characterizing the healthy vaginal microbiome throughout the entire lifecycle [14,15], as well as exploring the potential correlation between vaginal microbiome fluctuation and the development of uterine disease [16,17,18]. Antimicrobials remain a component of numerous treatment protocols employed on dairy cattle farms. The widespread use of antibiotics in dairy farming has given rise to several critical concerns about antimicrobial resistance and its effects on reproductive health. Commonly prescribed antibiotics can disrupt the balance of vaginal microbiota, possibly causing dysbiosis or with the potential to exacerbate existing dysbiosis. Recent studies have emphasized the necessity for targeted antimicrobial therapies that minimize adverse effects on commensal bacteria. In this context, probiotics and bacteriophages are being investigated as alternative approaches to modulate reproductive microbiota without contributing to resistance. Certain lactic acid bacteria strains, for instance, have demonstrated potential in reducing uterine infections and promoting microbial eubiosis [2]. Despite growing interest in this field, there are still substantial gaps in understanding the interplay between vaginal microbiota and uterine health. Research efforts should focus on characterizing the functional roles of specific bacterial species, identifying biomarkers of health and disease, and developing predictive models for uterine infections based on microbial profiles.

As a part of an innovation project aiming at the development of intravaginal probiotic preparation for dairy cows, this study aimed to contribute to the field by:Identifying the naturally occurring inhabitants of vaginal bacterial flora of Holstein-Friesian cows;Determining the sensitivity of selected, potentially probiotic bacterial strains to ten commonly used antibiotics;Establishing an average vaginal mucosa surface pH value for healthy Holstein Friesian cows.

## 2. Materials and Methods

### 2.1. Animals and Samplings

All animal handlings and samplings followed the applicable animal welfare rules (permission number: PE/EA/00775-4/2023 by the Pest County Government Office, Budapest, Hungary).

In total, 44 healthy multiparous Holstein Friesian cows on three large-scale Hungarian dairy cattle farms were involved in the study. Examinations and samplings were performed 7 days before the expected calving and 10.7 ± 6.4 after parturition. In total, 57 samplings were performed (36 before and 21 after calving), from which 13 cows were sampled before and after calving. Physical examination, including rectal temperature measurement, vaginoscopy, scoring vaginal discharge (VD) on a scale of 0–3 [8,19], body condition scoring on a scale of 1–5 [20], and locomotion scoring on a scale of 1–5 [21] was completed as a first step to ensure that the selected animals are healthy. Vaginal discharge was measured only in the cows after calving. During a rectal examination, the examiner tried to massage out some discharge from the vagina. VD was scored as follows: VD0 = clear or translucent mucus; VD1 = mucus containing flecks of white or off-white pus; VD2 = discharge containing ≤ 50% white or off-white mucopurulent material; and VD3 = discharge containing ≥ 50% purulent material, usually white or yellow, but occasionally sanguineous [8,19]. The vulva was cleaned with iodine solution and 70% alcohol spray, then vaginoscopy was performed for all cows with sterile single-use plastic Cusco vaginoscopes (Speculum S.A., Alfragide, Portugal) to assess cervical and vaginal appearance as well as the presence of purulent VD [22]. Only cows that did not have VD, or if they did have VD, it was scored as 0, and there was no visible sign of vaginitis, and no pus was present in the vagina were included in the study. After the vaginoscopy, vaginal pH levels were determined with a contact pH measurement device (Handheld PH20 pH meter, VWR International bvba., Leuven, Belgium) sterilized with 70% alcohol. Vaginal secretion samples were collected with a flushing method, using 100 mL/cow sterile phosphate-saline buffer. From the vaginal secretion samples, aerobic and anaerobic culturing was performed.

### 2.2. Bacterial Culturing

Sampling solution used: bovine serum albumin BSA (1.0 g/L); sodium chloride (8.5 g/L); L-cysteine (0.5 g/L); tween 80 (0.5 g/L) pH 7.4 +/− 0.2.

Ten-fold serial dilutions were made from each vaginal secretion sample in a physiological saline solution and were spread to nutrient agar, Columbia blood agar, thioglycolate agar, and MRS agar plates in two replicates. One plate of the replicate samples was cultured aerobically, and the other was cultured anaerobically at 38.5 °C. Each colony was cultivated further as a pure culture before MALDI-TOF analysis.

### 2.3. Matrix-Assisted Laser Desorption Ionization Time of Flight Mass Spectrometry (MALDI-TOF MS) Analysis

One bacterial colony was spread evenly on a spot of a 96-target polished steel BC plate, and then 1 μL of 70% formic acid (FA) solution was added. Following drying out, 1 μL of α-Cyano-4-hydroxycinnamic acid (α-CHCA) saturated matrix solution in a solvent (47.5% HPLC grade water, 2.5% trifluoroacetic acid, and 50% acetonitrile) was added. Two replicates were made from each bacterial sample. Afterwards, the target plate was loaded into the Microflex LT MALDI-TOF mass spectrometer (BRUKER Daltonics GmbH, Bremen, Germany). The *E. coli* DH5-alpha BRUKER Bacterial Test Standard, which covers the 3.6–17 kDa mass range, was applied to calibrate the system. The spectra were generated in linear positive ion mode. Mass spectra were acquired using flexControl 3.4 software. MBT Compass 4.1 software and MBT Compass Library were applied to identify the bacterial isolates.

### 2.4. Broth Microdilution Tests

The broth microdilution method outlined by CLSI [23] was utilized, employing 96-well sterile, flat-bottomed microplates for the assays. Five selected, potentially probiotic bacteria [12], *Brevibacillus agri* (*Bb. agri*), 2 strains of *Bacillus licheniformis* (*B. licheniformis*), *Bacillus pumilus* (*B. pumilus*), and *Bacillus subtilis* (*B. subtilis*), isolated from the vaginal microbiota of healthy dairy cows, were further examined. Brain heart infusion (BHI) broth containing the antimicrobial agent was dispensed into the microplate wells. The antimicrobial agents tested included amoxicillin trihydrate, ceftiofur hydrochloride, cefquinome sulphate, oxytetracycline hydrochloride, doxycycline hydrochloride, sulfamethoxazole, trimethoprim, florfenicol, marbofloxacin, tylosin tartrate, and tulathromycin. Control wells (0 mg/L) received 100 µL of BHI broth without antimicrobials. Next, bacterial inoculation was performed to achieve a final concentration of 10⁴ CFU/mL. The plates were then incubated at 37 °C for 24 h, after which bacterial growth was assessed by measuring absorbance at 600 nm using a microplate reader. The categories “S: sensitive”, “I: intermediate sensitive”, and “R: resistant” were determined based on MIC data, following CLSI guidelines [24,25].

### 2.5. Bacterial DNA Isolation and Polymerase Chain Reaction (PCR) Analysis

The five isolates were further examined with PCR. Before DNA isolation, bacterial broth cultures were grown for 24 h at 37 °C. Each 10 mL bacterial culture was centrifuged for 10 min at 3000 g. After removing the supernatant, gentle vortexing resuspended the cells in 1000 µL of sterile PBS. The DNA was extracted using the ZymoBIOMICS™ DNA Miniprep Kit (Zymo Research Corporation, Orange, CA, USA) according to the manufacturer’s instructions.

PCR analysis was applied to identify selected bacterial isolates and detect selected resistance genes taxonomically. We performed the PCR analyses on the CFX Opus Real-Time PCR System (BioRad Laboratories, Inc., Hercules, CA, USA). The applied primer sequences of the tested genes are given in Table 1. The final reaction volume of 20 μL contained 0.2 μM of the corresponding primers, 1× concentrated SsoAdvanced Universal SYBR Green Supermix (BioRad Laboratories, Inc., Hercules, CA, USA) in nuclease-free water, and the 2 μL 2 ng/μL DNA sample, which was added directly to the PCR reaction mixture for each PCR reaction. PCR reactions were run at a thermal cycle of 95 °C for 3 min, followed by 40 cycles at 95 °C for 20 s, then at 60 °C for 30 s, and then at 72 °C for 30 s. At the end of each cycle, there was a 10-second-long fluorescence monitoring.

## 3. Results

The cows involved in the study exhibited no indications of disease upon the pre-sampling examination, and they did not show vaginal discharge either pre- or post-calving. The average pH of the vaginal mucosa surface was 7.15 ± 0.17 before calving and 7.20 ± 0.27 post-calving (raw data are in Appendix A). A t-test showed no significant difference between the two.

A total of 54 bacterial species were identified in the vaginal samples of healthy cows. The predominant bacterial category was Gram-positive, accounting for 47 species (87%), while Gram-negative bacteria constituted 7 species (13%). The results of the species identification are shown in Table 2.

The frequency of each genus of the identified bacteria is presented in Figure 1. Out of the 47 Gram-positive species, 13 are classified as *Bacillus*, making it the most prevalent genus within healthy vaginal bacterial biota. It is followed by the *Streptococcus* genus, with 10 out of 47 species. *Staphylococcus* is the third most prevalent genus, with 6 out of 47 species. The genera *Brevibacillus*, *Corynebacterium*, and *Paenibacillus* are each represented by three species. The remaining Gram-positive and all Gram-negative genera are represented by a single species each.

The taxonomic classification of the five selected, putative probiotic bacterial isolates (*Bb. agri*, 2 *B. licheniformis strains*, *B. pumilus*, and *B. subtilis)* was verified by PCR. Subsequently, resistance genes were examined with PCR, and the results are presented in Table 3. *Brevibacillus agri* and *B. licheniformis* genomes did not contain beta-lactamase resistance genes, while the other three isolates did.

The minimum inhibitory concentration (MIC) of antimicrobial substances against the five isolates was determined by the 96-well plate broth dilution method. The results are shown in Table 4 and Table 5, and the raw data are in Appendix A. Ten different antibiotics were selected for this examination: amoxicillin, cefquinome, ceftiofur, doxycycline, florfenicol, marbofloxacin, oxytetracycline, trimethoprim-sulfamethoxazole, tulathromycin, and tylosin. It was observed that each tested isolate exhibited resistance to at least one antibiotic, but they showed sensitivity to most of the tested agents, as per CLSI guidelines [24,25].

## 4. Discussion

This study provides insights into the composition of the vaginal microbiota in healthy Holstein Friesian cows and its antibiotic susceptibility patterns. Our findings contribute to the growing body of research highlighting the pivotal role of microbiota in bovine reproductive health, particularly in maintaining a balanced microbial environment that may prevent uterine diseases in dairy cows.

The average vaginal mucosa pH value of 7.17 ± 0.21 observed in this study is consistent with the pH levels previously reported in cattle [27]. The moderately alkaline, near-neutral pH is consistent with the low abundance of *Lactobacillus* spp. As this genus is capable of producing significant quantities of lactate metabolites, it frequently results in low vaginal pH in other species, including primates and humans [28]. Beckwith-Cohen et al. [29] reported a range of 5.52–8.60 for cattle vaginal pH and found that the average vaginal pH of Israeli Holstein multiparous cows was 7.35. The study revealed that, regardless of the prenatal period, the vaginal pH of cows and heifers exhibits greater variability compared to that of first-calf heifers, with vaginal pH values tending to be more acidic in cows. The pH regulation is critical in microbial colonization and stability, influencing the abundance and viability of beneficial bacteria. Maintaining this pH range in healthy cows may be essential for assessing reproductive tract health and predicting infection susceptibility.

The identification of 54 bacterial species in the vaginal microbiota of healthy cows, with a predominant presence of Gram-positive bacteria (87%), aligns with previous studies describing a similar microbial composition [11,12,15,18]. The dominance of *Bacillus*, *Streptococcus*, and *Staphylococcus* species further supports the hypothesis that these bacterial genera contribute to the maintenance of a stable vaginal environment. Notably, *Bacillus* species, comprising the most identified bacteria, have been suggested to possess probiotic properties, potentially contributing to microbial homeostasis and pathogen resistance [12,30].

Five strains were selected (*Bb. agri*, *B. licheniformis* (W), *B. licheniformis*, *B. pumilus*, and *B. subtilis*) for further investigations into antibiotic resistance. These non-pathogenic strains are generally regarded as putative probiotic strains [12,27,30,31,32,33,34,35,36] and can be potentially used in intravaginal probiotic preparations. During susceptibility tests, the beta-lactamase resistance gene did not invariably lead to resistance to beta-lactam antibiotics. Bacteria carrying the class A beta-lactamase-encoding gene exhibited a high resistance to amoxicillin. However, those holding the class D beta-lactamase-encoding gene did not consistently resist the tested beta-lactam compounds. The observed resistance to beta-lactam antibiotics in certain isolates is a matter of concern, as it underscores the potential impact of antibiotic use in dairy farming on selecting resistant strains [37]. Nevertheless, the overall level of susceptibility of the isolates to the majority of the antibiotics indicates that antimicrobial treatments may not have a substantial effect on commensal bacteria in controlled conditions. The presence of resistance genes in specific isolates, particularly those encoding beta-lactamase and tetracycline resistance, requires further investigation to determine their potential for horizontal gene transfer among the bovine reproductive tract microbiota [16,17].

The findings of this study have significant implications for the management of dairy herds and the implementation of reproductive health strategies. A comprehensive understanding of the composition of the vaginal microbiota, in conjunction with its interaction with antimicrobial agents, can facilitate the development of targeted interventions that are aimed at preserving microbial balance. The potential role of probiotics in supporting stable microbiota and mitigating the effects of antibiotic-induced dysbiosis is an avenue for future research [32,38].

Some limitations of the study should be acknowledged. The sampling was conducted on a relatively small number of dairy farms, and environmental or management factors may have influenced microbiota composition. Additionally, while culture-based methods and PCR analyses were employed for bacterial identification and resistance gene detection, metagenomic approaches have the potential to provide a more comprehensive assessment of microbial diversity and functional capabilities. Future research should focus on longitudinal studies to track microbiota changes throughout the reproductive cycle and their potential association with reproductive performance and disease resistance.

## 5. Conclusions

In conclusion, this study enhances our understanding of the vaginal microbiota in dairy cows and its susceptibility to commonly used antibiotics. The findings highlight the significance of microbial balance in reproductive health and underscore the need for the prudent use of antibiotics in dairy farming. Further investigations into the functional roles of specific bacterial species and the potential benefits of probiotic supplementation may pave the way for novel strategies to improve reproductive outcomes in dairy cattle.

## Figures and Tables

**Figure 1 vetsci-12-00423-f001:**
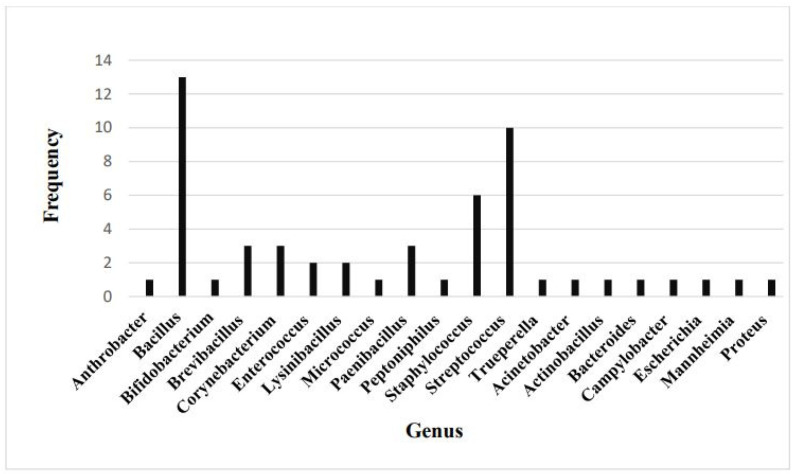
Frequency of bacterial genera present in the normal vaginal flora of healthy dairy cows.

**Table 1 vetsci-12-00423-t001:** Sequence of primer sets used for PCR analyses.

Species	Gene Symbol	Gene	Accession Number		Primer	Product Size (Base Pair)
*Brevibacillus*	*penP*	beta-lactamase class A	BA6348_08610	F	CTGTGCAAAATGGCGCGTAT	168
*agri*				R	CCGATCCTCCCACCAAATCC	
	*ITS*	internal transcribed spacer	AF478091.1	F	CCCGAAGTCGGTGAGGTAAC	214
				R	ATGGACGCGAGTGCTCTTAG	
*Bacillus*	*penP*	beta-lactamase class A	BLi00280	F	GCAATCACTCGAATGCCTCAC	178
*licheniformis*				R	ATCGTCGATGCAAAAGCGAAG	
	*ITS*	internal transcribed spacer	AF478086.1	F	ATGCCGCGGTGAATACGTTC	161
				R	CACCTTCCGATACGGCTACC	
*B. pumilus*	*tetA_3*	major facilitator superfamily (MFS)	SAMEA4076707	F	ATTGTCGGACCGAGCCTTG	141
		transporter	_03279	R	AGAAACTGTCGAAGGATGCTG	
	*bpu-1*	beta-lactamase class D	BPUM_RS11670	F	GAAGAGAAACACGCCACCCT	124
				R	TGCCGGTGCCTTTGATATTTG	
	*ITS*	internal transcribed spacer	AF478070.1	F	TATATGGAGCAGCGTGCGTT	226
				R	CATCGGCTCCTAGTGCCAAG	
*B. subtilis*	*penP*	beta-lactamase class A	BSU_18800	F	TCTCACGACTGACAAACGCA	122
				R	TTCCGGCTCCGGATTTATCG	
	*bsu-2*	beta-lactamase class D	BSU_02090	F	AGTTTTGGCTGCAAAGCTCG	168
				R	TTCCGGTTTTCCCGTAGAGC	
	*ITS*	internal transcribed spacer	AB050658.1	F	ACAGAACGTTCCCTGTCTTGT	124
				R	TCACTACGTGATATCTTGCATTACT	

All primers were designed applying the Primer-BLAST tool (http://www.ncbi.nlm.nih.gov/tools/primer-blast, accessed on 29 September 2023) [26], F—forward, R—reverse.

**Table 2 vetsci-12-00423-t002:** Bacterial species present in the normal vaginal microbiota of healthy dairy cows.

Aerobic/Anaerobic Characteristics	Gram-Positive	Gram-Negative
Aerobe	*Bacillus circulans*, *B. oceanisediminis*, *B. oleronius*, *B. pumilus*, *B. safensis* *B. sonorensis*, *Brevibacillus agri/parabrevis*, *Bb. borstelensis*, *Corynebacterium xerosis* *Lysinibacillus fusiformis*, *L. massiliensis*, *Bacillus amyloliquefaciens*, *B. clausii*, *B. megaterium*, *B. siralis*, *Arthrobacter gandavensis*, *Micrococcus luteus*	*Acinetobacter pittii*
Facultative anaerobe	*Bacillus cereus*, *B. licheniformis*, *B. subtilis*, *Corynebacterium camporealensis*, *C. renale*, *Enterococcus avium*, *E. hirae*, *Paenibacillus cookii*, *P. ihumii*, *P. lactis*, *Staphylococcus chromogenes*, *S. epidermidis*, *S. hominis*, *S. succinus*, *S. sciuri*, *S. xylosus*, *Streptococcus alactolyticus/lutetiensis*, *S. canis*, *S. dysgalactiae*, *S. equinus*, *S. pluranimalium/hyovaginalis*, *S. pneumoniae/pseudopneumoniae*, *S. suis*, *S. uberis*, *S. mitis/oralis/peroris*, *Trueperella pyogenes*	*Actinobacillus rossii*, *Escherichia coli*, *Mannheimia varigena*/*haemolytica*/*granulomatis*, *Proteus mirabilis*
Obligate anaerobe	*Bifidobacterium pseudolongum*, *Peptoniphilus indolicus*	*Bacteroides fragilis*
Microaerophile		*Campylobacter hyointestinalis*

**Table 3 vetsci-12-00423-t003:** The presence of the investigated resistance genes in the five selected bacterial isolates.

Bacterial Isolate	Resistance Gene	Present
*Brevibacillus agri*	Beta-lactamase class A	No
*Bacillus licheniformis*	Beta-lactamase class A	Yes
*Bacillus licheniformis* (W)	Beta-lactamase class A	No
*Bacillus pumilus*	Beta-lactamase class D	Yes
	Tetracycline resistance MFS efflux pump	Yes
*Bacillus subtilis*	Beta-lactamase class A	Yes
	Beta-lactamase class D	Yes

**Table 4 vetsci-12-00423-t004:** Minimum inhibitory concentration (MIC—mg/L) of the tested antimicrobial agents.

	AMX	CFQ	CTF	DOX	FFC	MBF	OTC	T-S	TUL	TYL
*Brevibacillus agri*	0.5	0.06	0.016	0.05	2	0.5	1	500	0.5	0.5
*Bacillus licheniformis*	>50	1	0.5	0.5	4	0.125	2	1	0.5	0.5
*B. licheniformis* (W)	>5	>8	8	>0.05	4	0.25	0.5	2	2	0.25
*B. pumilus*	>5	>8	>2	0.05	4	0.5	0.5	>2	1	0.5
*B. subtilis*	>50	0.25	0.25	0.5	1	0.25	8	1	2	0.5

AMX—amoxicillin; CFQ—cefquinome; CTF—ceftiofur; DOX—doxycycline; FFC—florfenicol; MBF—marbofloxacin, OTC—oxytetracycline; T-S—trimethoprim-sulfamethoxazole; TUL—tulathromycin; TYL—tylosin.

**Table 5 vetsci-12-00423-t005:** Sensitivity of the selected isolates to ten different antibiotics.

	AMX	CFQ	CTF	DOX	FFC	MBF	OTC	T-S	TUL	TYL
*Brevibacillus agri*	S	S	S	S	S	S	S	R	S	S
*Bacillus licheniformis*	R	S	S	S	S	S	S	S	S	S
*B. licheniformis* (W)	I	R	R	S	S	S	S	S	S	S
*B. pumilus*	I	R	I	S	S	S	S	I	S	S
*B. subtilis*	R	S	S	S	S	S	R	S	S	S

S—sensitive, I—intermediate sensitive; R—resistant; AMX—amoxicillin; CFQ—cefquinome; CTF—ceftiofur; DOX—doxycycline; FFC—florfenicol; MBF—marbofloxacin, OTC—oxytetracycline; T-S—trimethoprim-sulfamethoxazole; TUL—tulathromycin; TYL—tylosin.

## Data Availability

The data presented in this study are available on request from the corresponding author.

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
