# Peer review of "Identification of Naturally Occurring Inhabitants of Vaginal Microbiota in Cows and Determination of Their Antibiotic Sensitivity"

_vetsci, 2025, doi:10.3390/vetsci12050423_

Round 1
Reviewer 1 Report
Comments and Suggestions for Authors
The manuscript provides reference and insights into the role of microbial communities in dairy cow reproductive health and the use of antibiotics in breeding management through bacterial isolation, identification, and antibiotic susceptibility testing. However, there are issues with insufficient rigor. It is recommended to make some revisions.
1. Section 2.1 lacks details. Lines 96-97 only mention "7-14 days before and after calving" but do not specify the exact sampling time points or whether multiple samplings were conducted. It is recommended to add a supplementary table explaining the specific sampling time points and distribution information.
2. "Table 1" (Line 177) should indicate the aerobic/anaerobic characteristics of the isolated strains.
3. Duplicate numbering issue. There are two "Table 1" instances in the paper (Line 166 and Line 177)—one for PCR primer sequence information and the other for bacterial species in the vaginal microbiota of dairy cows. It is recommended to modify the numbering of one of the tables.
4. Results section ("Five bacterial," Line 189) is vague. It does not explain why "Brevibacillus" is was selected for drug resistance gene screening instead of more abundant genera like "streptococcus" and "Staphylococcus" in "Figure 1." A clarification should be added.
5. Results section (P171) lacks statistical analysis of pH changes before and after calving. It is recommended to supplement this part.
Author Response
Rev.: The manuscript provides reference and insights into the role of microbial communities in dairy cow reproductive health and the use of antibiotics in breeding management through bacterial isolation, identification, and antibiotic susceptibility testing. However, there are issues with insufficient rigor. It is recommended to make some revisions.
AU: Thank you for your valuable comments and suggestions. We have tried to improve the quality of the manuscripts.
Rev.: 1. Section 2.1 lacks details. Lines 96-97 only mention "7-14 days before and after calving" but do not specify the exact sampling time points or whether multiple samplings were conducted. It is recommended to add a supplementary table explaining the specific sampling time points and distribution information.
AU: We found an extra table unnecessary. The text was supplemented with the following sentence: Examinations and samplings were performed 7 days before the expected calving and 10.7±6.4 after parturition.
Rev.: 2. "Table 1" (Line 177) should indicate the aerobic/anaerobic characteristics of the isolated strains.
AU: Table 2 has been modified. The oxygen requirements of the bacteria are included in the table.
Rev.: 3. Duplicate numbering issue. There are two "Table 1" instances in the paper (Line 166 and Line 177)—one for PCR primer sequence information and the other for bacterial species in the vaginal microbiota of dairy cows. It is recommended to modify the numbering of one of the tables.
AU: It is corrected.
Rev.: 4. Results section ("Five bacterial," Line 189) is vague. It does not explain why "Brevibacillus" was selected for drug resistance gene screening instead of more abundant genera like "streptococcus" and "Staphylococcus" in "Figure 1." A clarification should be added.
AU: Our goal was to select bacteria from the cows' vaginal microbiota with potential probiotic properties, and we aimed to identify the resistance status of those bacteria to evaluate if they can be safely used as intravaginal probiotic preparations. Therefore, we have not screened antimicrobial resistance genes in streptococci and staphylococci, which were not probiotic candidates. The text has been clarified in the introduction, materials, and results sections.
Rev.: 5. Results section (P171) lacks statistical analysis of pH changes before and after calving. It is recommended to supplement this part.
AU: It is supplemented. There was no difference between the two values.
Reviewer 2 Report
Comments and Suggestions for Authors
We are still beginning to understand the role of the vaginal microbiota in cow health and fertility, and this study helps solidify the foundations on which future studies will be based.
The number of animals used is relatively small and geographically confined, but even so, the results help to define trends.
However, there are several aspects that could be improved.
Was everything written from line 43 to 51, such as e.g. "estrogen levels have been shown to promote the proliferation of specific bacterial species during the oestrous cycle" stated in references #2 and #6?
In lines 68 and 69, the authors state that "commonly prescribed antibiotics can disrupt the balance of vaginal microbiota, with the potential to exacerbate existing dysbiosis." The possibility of them causing dysbiosis themselves must also be considered.
In line 99, the authors report that they performed “metritis scoring on a scale of 0-3”. Despite what is technically described in the references (#8, #19), what was done specifically in this study is not very clear, which raises some concerns as it would be essential to ensure that the animals were as healthy as possible, so that the vaginal flora would not be influenced by a concomitant infection and the respective immune response.
Therefore, I ask what was performed to diagnose possible subclinical conditions? Vaginoscopy to assess cervical and vaginal appearance as well as the presence of purulent vaginal discharge? Transrectal ultrasonographic examination? And cytological sampling, namely polymorphonuclear neutrophil counts? If using only one method, cytology would perhaps be the most suitable for this study.
The methodologies described in the following study, e.g., could be useful to obtain more reliable results, since the presence of subclinical infections can alter the population patterns of the vaginal microbiota:
Bazzazan, A., Vallejo-Timaran, D. A., Maldonado-Estrada, J., Segura, M., & Lefebvre, R. (2024). Diagnosis of clinical cervicitis and vaginitis in dairy cows in relation to various postpartum uterine disorders. Clinical Theriogenology, 16.
Author Response
Rev.: We are still beginning to understand the role of the vaginal microbiota in cow health and fertility, and this study helps solidify the foundations on which future studies will be based. The number of animals used is relatively small and geographically confined, but even so, the results help to define trends. However, there are several aspects that could be improved
AU: Thank you for your valuable review and your suggestions. We have clarified the text accordingly.
Rev.: Was everything written from line 43 to 51, such as e.g. "estrogen levels have been shown to promote the proliferation of specific bacterial species during the oestrous cycle" stated in references #2 and #6?
AU: Yes, mainly. In this section, we clarified the text with more references.
Rev.: In lines 68 and 69, the authors state that "commonly prescribed antibiotics can disrupt the balance of vaginal microbiota, with the potential to exacerbate existing dysbiosis." The possibility of them causing dysbiosis themselves must also be considered.
AU: The text is supplemented accordingly.
Rev.: In line 99, the authors report that they performed "metritis scoring on a scale of 0-3". Despite what is technically described in the references (#8, #19), what was done specifically in this study is not very clear, which raises some concerns as it would be essential to ensure that the animals were as healthy as possible, so that the vaginal flora would not be influenced by a concomitant infection and the respective immune response. Therefore, I ask what was performed to diagnose possible subclinical conditions? Vaginoscopy to assess cervical and vaginal appearance as well as the presence of purulent vaginal discharge? Transrectal ultrasonographic examination? And cytological sampling, namely polymorphonuclear neutrophil counts? If using only one method, cytology would perhaps be the most suitable for this study.
AU: We performed vaginoscopy. The text is supplemented with this and also with the description of the measurements of vaginal discharge.
Rev.: The methodologies described in the following study, e.g., could be useful to obtain more reliable results, since the presence of subclinical infections can alter the population patterns of the vaginal microbiota:
Bazzazan, A., Vallejo-Timaran, D. A., Maldonado-Estrada, J., Segura, M., & Lefebvre, R. (2024). Diagnosis of clinical cervicitis and vaginitis in dairy cows in relation to various postpartum uterine disorders. Clinical Theriogenology, 16.
AU: Thank you. We used this study as a reference.
Reviewer 3 Report
Comments and Suggestions for Authors
The present study is very interesting and important. The vaginal microbiome is a key component of reproductive health in dairy cattle, as it is the first line of defence against external pathogens and ensures a healthy uterine environment. The manuscript is written in a clear manner. Purpose of the work was precisely defined.
Minor corrections:
Latin names of microorganisms should be given in full the first time they are used and their abbreviation e.g. Bacillus licheniformis -italics (B. licheniformis – italics ) and then only the abbreviation should be used throughout the text.
Line 81: rather should be: „vaginal microbiota” than „vaginal bacterial flora”
Author Response
Rev.: The present study is very interesting and important. The vaginal microbiome is a key component of reproductive health in dairy cattle, as it is the first line of defence against external pathogens and ensures a healthy uterine environment. The manuscript is written in a clear manner. Purpose of the work was precisely defined.
AU: Thank you for your valuable review and your suggestions.
Minor corrections:
Rev.: Latin names of microorganisms should be given in full the first time they are used and their abbreviation e.g. Bacillus licheniformis -italics (B. licheniformis – italics ) and then only the abbreviation should be used throughout the text.
AU: Thank you. We applied your suggestion in the main text. The tables should be understandable and clear without reading the text; therefore, we wrote the bacteria names in full. Or, if there were more names from the same genus in a table, we wrote the first mention of that genus in full, then abbreviated (see Table 2, for instance).
Rev.: Line 81: rather should be: "vaginal microbiota" than "vaginal bacterial flora"
AU: Thank you for pointing this out. We respectfully keep the original text. Microbiota includes bacteria, archaea, protists, fungi, and viruses, but we only dealt with bacteria.